# Opportunities for Recycling PV Glass and Coal Fly Ash into Zeolite Materials Used for Removal of Heavy Metals (Cd, Cu, Pb) from Wastewater

**DOI:** 10.3390/ma16010239

**Published:** 2022-12-27

**Authors:** Maria Visa, Alexandru Enesca

**Affiliations:** Product Design, Mechatronics and Environmental Department, Transilvania University of Brasov, Eroilor 29 Street, 35000 Brasov, Romania

**Keywords:** PV glass, alkaline coal fly ash, hydrothermal method, adsorption, heavy metals

## Abstract

This work shows the development and characterization of two zeolite structures by recycling PV glass and coal fly ash for the removal of cadmium, copper, and lead from synthetic solutions containing one or three cations. The materials were characterized in terms of crystalline structure (XRD), morphology (SEM, AFM), and specific surface. For increasing the heavy-metals removal efficiency, the adsorption conditions, such as substrate dosage, preliminary concentration, and contact time, were optimized. The pseudo-second-order kinetic model adsorption kinetics fit well to describe the activity of the zeolites ZFAGPV-A and ZFAGPV-S. The zeolite adsorption equilibrium data were expressed using Langmuir and Freundlich models. The highest adsorption capacities of the ZFAGPV-A zeolite are q_maxCd_ = 55.56 mg/g, q_maxCu_ = 60.11 mg/g, q_maxPb_ = 175.44 mg/g, and of ZFAGPV-S, are q_maxCd_ = 33.45 mg/g, q_maxCu_ = 54.95 mg/g, q_maxPb_ = 158.73 mg/g, respectively. This study demonstrated a new opportunity for waste recycling for applications in removing toxic heavy metals from wastewater.

## 1. Introduction

The issue of sustainable development should be solved by finding new technologies and new processes whose environmental impact would be limited. Water resources are limited, and water is the most affected actor from the environment, serving as a pollutant collector for different contaminants, such as heavy metals, dyes, surfactants, fats, and oils [1]. Detergents, heavy metals, and anionic or cationic dyes are taken into the body by inhalation, ingestion, or skin absorption. Heavy metal ions, such as Cd^2+^, Cu^2+^, and Pb^2+^ in aquatic systems, are absorbed by living organisms, cannot be metabolized, and pass up in the food chain to humans [2,3]. The presence of hydrated metallic ions is considered a higher hazard risk compared with metal atoms because the absorption occurs faster, increasing the enzymatic stress processes and constituting a potential hazard to human health. 

Exposure to cadmium generally occurs through food, as it is bioaccumulated by plants. Vegetables and cereals often contain large amounts of cadmium [4,5]. The copper in drinking water is usually at 5 × 10^−2^ mg L^−1^ and can be extended in some circumstances up to 0.1 mg L^−1^. Lead is a persistent pollutant and is considered among the highest toxic heavy metals discharged into the environment [6,7]. The originating factors of heavy metals in surface water, underground water, wastewater, soil, and airborne dust are [8] electroplating and metal surface treatment, fusible alloys metallurgy, petroleum refineries, the smelting of metals, Ni-Cd-based rechargeable battery development, lead-acid batteries, electronic factories, leather dyes and pigments, mining activities, the production of glass and ceramics, fertilizers, and pesticides [8,9]. Saving environmental factors, such as flora and fauna, and avoiding health problems, imposing strict limits on pollutant discharge and modern wastewater treatment processes eventually make treated water ready for reuse. 

Several techniques can be involved in heavy metals removal: coagulation and flocculation [10,11,12], reagents used for chemical precipitation to obtain heavy metal compounds with low solubility (carbonates or hydroxides), biosorption [13], electrodialysis [14], reverse osmosis [15], solvent extraction, ion exchange [16], phytoremediation (phytoextraction) [5], adsorption [17], and biosorption [18].

Compared to other technology, adsorption can be an alternative using natural substrates, such as bentonite, clay, diatomite [19,20,21], magadiite materials [22], or activated carbon obtained from agriculture and forest waste, such as dried banana peels [23], orange peel [24], wheat straw [25], wood [26] and coffee grounds [27]. The major advantages of adsorption are related to which adsorbent materials synthesis technology is accessible. 

Coal fly ash resulting as a byproduct of coal combustion is a potential adsorbent substrate for heavy metals removal from wastewater. A small amount of coal fly ash is used in building materials, such as cement, while the rest is disposed of in landfills, leading to the pollution of air, groundwater, and soil. Coal fly ash has a negative impact on the surrounding environment which cannot be completely removed. One efficient solution is to transform coal fly ash and oil shale [28] into zeolite, such as zeolite-A [29,30], Na-P1 [18], zeolite-X [31], K-chabazite and K-phillipsite [32], mesoporous silica materials (MCM-41) [33] by various methods. Feldspars and zeolite K-F were obtained from waste container glass [34]. 

The composition of coal fly ash, mainly the SiO_2_/Al_2_O_3_ ratio, is very important for obtaining a zeolite. To increase this ratio, in friendly conditions, one cheap alternative is to use the glass from photovoltaic panels (G-PV) after operating life. Depending on the quality of materials, a photovoltaic panel service life can extend up to 25 years [35]. Therefore, the quantities of waste obtained from broken PV panel waste can reach 1,957,099 t by 2038 [36], and disposal of photovoltaic systems at the end of the operating period is a major problem for the environment and human health. Recycling all components of PV panels is a necessity. There are studies that report on PV waste removal based on thermal high-energy consumption treatments that are able to eliminate ethylene-vinyl acetate (EVA) foil [37]. The literature includes papers reporting recycling procedures for silicon-based photovoltaic modules by including them in waste bath polymeric electric blends in order to obtain new composite materials [38]. The thermal stability, chemical resistance, catalytic properties, ion exchange, and adsorption capacity determine the many applications of zeolites [39]. 

This work describes the development of new zeolite materials based on two wastes: coal fly ash and glass extracted from broken photovoltaic silicon panels. These materials are characterized in terms of composition and morphology as well as their properties to efficiently eliminate heavy metals from wastewater.

## 2. Materials and Methods

### 2.1. Raw Compounds

The raw coal fly ash used was locally extracted from a thermoelectric plant (CET Brasov, Romania). Using coal fly ash as a heterogeneous compound is based on the different melting temperatures of its components. The grains are characterized by spherical shape and various sizes (from 5–300 μm) obtained from the electrofilters. The metal oxide composition (SiO_2_, Al_2_O_3_, and Fe_2_O_3_) covered 70% of the total mass, and the SiO_2_/Al_2_O_3_ ratio (2.41) showed that coal fly ash represents a precursor for zeolites materials [18,40].

The glass obtained from photovoltaic panels (PV) after detaching the subcomponents (junction box, aluminum frame, wires) was a new material added to synthesis of zeolites together with coal fly ash (FA). The glass collected was washed, dried, and ground with the ZM 200 Retsch (ZM 200 model, Berlin, Germany) grinding mill. The grinding speed of the glass was 8000 rpm. The fragments resulting from grinding for one minute had dimensions between 130–260 µm.

### 2.2. Zeolite Materials Synthesis and Characterization

The coal fly ash fraction (40 µm) and PV glass (1:1) ratio were washed in ultrapure water (FA+G-PV: water (100:1, g:L) under mechanical stirring at ambient temperature (20–21 °C) for 24 h then filtrated and dried at 115 °C for 24 h. The mixture (FA+G-PV) and solid NaOH were submitted to alkaline fusion to obtain the intermediates (MAFPGV). The second step represented by hydrothermal treatment was required to transform the intermediates into zeolite materials. The alkali-coal fly ash fusion induced the dissolution of Si^4+^ from SiO_2_ and of Al^3+^ from γ-Al_2_O_3_ with the development of easily soluble silicates and Na-aluminates and -silicates, enhancing zeolite material formation [41]. Under these conditions, the condensation of Na-silicate and -aluminate forms the aluminosilicate gel which then crystallizes [42,43]. The traditional hydrothermal method for alkaline FA activation has been reported in other studies [18,44]. The zeolites were developed by combining the hydrothermal and alkaline fusion treatment of the coal fly ash and PV glass with different amounts of NaOH, inducing the formation of analcime, sodalite, chabazite K, ZSM-5(H), zeolite N, cancrinite, and tobermorite. 

Details for the precursor alkaline fusion and hydrothermal treatment are provided in Figure 1. The hydrothermal setup had the following parts: 500 mL reactor, vacuum pump, internal liquid circuit, and heating internal circuit. The samples were obtained within 5 h at 150 °C temperature and 5 atm pressure. The resulting zeolites were denoted ZFAGPV-S and ZFAGPV-A. The main difference between the two zeolites consists of the hydrothermal mixture composition: MAFPGV mixed with water and NaOH was used for ZFAGPV-S development, and MAFPGV mixed with water was employed for ZFAGPV-A formation.

During the alkaline fusion process, the PV glass and the quartz found in the washed coal fly ash (FA_w_) were submitted to a partial or total dissolution, function of the process temperature. The temperature parameter (500–600 °C) is very important in alkaline fusion reaction of SiO_2_ from PV glass, as presented in Equation (1).
SiO_2_ + 4NaOH → Na_4_SiO_4_ + H_2_O,(1)

The literature includes articles reporting various methods of zeolite development starting from coal fly ash, such as modifying temperature, FA/NaOH or FA/KOH ratio, gel stirring time, or crystallization time [32,43,45,46,47]. These papers represent an alternative for zeolite formation and include optimized physical parameters.

### 2.3. Material Characterization

Considering the correlation of precursor physical-chemical properties and the zeolite materials with adsorption activity, these materials were characterized. The crystallinity was studied with a Bruker diffractometer (XRD, Bruker D8 Model, Berlin, Germany), and the parameters were 1.5406 Å (Kα1), 0.01 step size, 40 kW, 40mA, and scan speed 1 s/step with 2θ = 10 up to 80°. EDX analysis (Thermo, Waltham, MA, USA) and Fourier transform infrared spectroscopy (FTIR, Spectrum BX model, PerkinElmer, Waltham, MA, USA) were used as well. The morphology was investigated by atomic force microscopy (AFM, NT-NDM model, Amsterdam, Netherlands) in semicontact regime with Si-based tips with a radius of 10 nm by applying a constant force of 0.15 N/m and scanning electron microscopy (SEM Hitachi model S-3400 N type 121 II, Tokyo, Japan). BET-specific surface and porosity analyses were measured using an Autosorb-IQ-MP (Quantachrome Instruments, Boynton Beach, FL, USA), and the static contact angle was evaluated with the sessile drop method (OCA-20 ContactAngle-meter, DataPhysics Instruments, Berlin, Germany).

## 3. Results

### 3.1. Crystalline Structure of Substrates

The XRD analysis of the FA and PV glass employed for the zeolite development is presented in Figure 1a. In accordance with X-ray diffraction results, the zeolites do not consist of a pure phase or glass quartz (SiO_2_) phase, and γ-Al_2_O_3_ with hematite and ramsdellite (Mn_2_O_3_) can be encountered while SiO_2_ is the main oxide in the PV glass. The zeolite samples exhibit polycrystalline structures, and the percentage of crystallinity varies from 49.91% (ZFAGPV-S) to 61.82% (ZFAGPV-A) and 68.70% for MFAPVG after alkaline fusion.

Analcime, sodalite, and aluminosilicates crystal size values were calculated using the Scherrer equation and are 145–277 Å for analcime, 417–440 Å for analcime syn, and 383–397 Å for sodalite and sodium aluminum silicate hydrate, respectively. The crystallite size can be influenced by NaOH dosage in hydrothermal treatment.

The XRD diffractogram in Figure 1b shows predominant crystalline components of ZFAPVG-S: sodium aluminum silicate hydrate (Na_6_(AlSiO_4_)_6_·4H_2_O), sodalite (Na_8_AlSiO_4_)_6_(OH)_2_·4H_2_O, cancrinite (Na_6_Al_6_Si_6_O_24_)(OH)_2_·(H_2_O)_6_ (101) with hexagonal structure of the crystallite, and nonstoichiometric zeolite N (K_11.5_Na_0.5_Al_10_Si_100_O_4_ (OH)_2_·x(H_2_O). The diffraction pattern of the ZFAPVG-A sample shows numerous peaks corresponding to analcime (Na(AlSi_2_O_6_)·H_2_O), analcime syn (Na_0.931_(AlSi_2_O_6_)·H_2_O), and chabazite-Na (NaAlSi_2_O_6_·3H_2_O) with a rhombohedral H axe structure of the crystallites.

The diffractogram patent of MFAPVG shows that quartz (SiO_2_) from FA_w_ or PV glass is still untransformed, but some of them formed nepheline syn (syn = syenite) (NaAlSiO_4_), nepheline potassium syn (K,Na)AlSiO_4_, and muscovite -2M1 sodian syn (Na_2_K_2_(Al_12_Si_12_O_40_)(OH)_8_. The peaks at Ɵ = 33.94° and Ɵ = 38.64° show sodium iron oxide (Na_2_FeO_3_) and maghenite –Q syn (Fe_2_O_3_) respectively.

### 3.2. Morphological and Surface Energy Properties

#### 3.2.1. Surface Morphology

SEM analysis provides information related to particle size and shape and the aggregated formation from the coal fly ash and PV glass (Figure 2a insets). This type of evaluation may consider random particles identified during the analysis and may not include other relevant entities. The SEM images of the FA powder indicate the presence of spherical grains with sizes varying from 19.8 to 41.4 µm. The glass beads in FA have been converted into other particle shapes during the zeolite development. The images in Figure 2a,b shows aggregates of different sizes and shapes but also the dispersed spherical nanoparticles. The alkaline fusion and hydrothermal processes developed in the autoclave between the FA, PV glass, and NaOH included chemo-physical property modifications such as changes to the chemical composition and morphology of the surface. Shirani Lapari reported sodalite with regular spherical morphology which was synthesized under a hydrothermal process from aluminum-silicate aggregates obtained from a NaOH solution, NaAlO_2,_ and fumed silica [48].

The analcime particles with sizes of 920 nm can combine and form agglomerates with sizes between 6.30–33.2 µm. The analcime EDS analysis indicates that the atomic ratio between Na:Si:Al is 1.1:1.9:1.0, consisting of the Si/Al ratio range corresponding to the zeolite analcime [49].

The 2D and 3D AFM images (Figure 3a,c) can be employed to obtain information on pore size distribution using the method described by [50,51] (Figure 3b,d). The AFM images (5 × 5 µm scan area) provide a clue regarding the roughness and pores/voids distribution. The analysis was made from a different point, and the figures include the most representative results. It must be underlined that the numerical values are averages of the three analysis points. The porosity and surface roughness of the materials are different when different quantities of NaOH were added.

This evaluation indicates that the aggregates were formed from medium-sized particles (<1 µm). Due to the particles’ random distribution into the aggregates, the average roughness varies from 20.527 nm to 27.149 nm, respectively. The macropore evolution curve with more than one maximum confirms the random aggregation. Thus, the low roughness ZFAGPV-A value shows a higher number of aggregates with different geometries (mostly round). The quartz conversion to aluminosilicate compounds is confirmed by the newly shaped grains corresponding to zeolite agglomerates (Figure 2 and Figure 3). 

The porosity properties were studied on powder samples based on the adsorption-desorption procedure. The pores’ volumes and diameters as well as the specific surface area (S_BET_) were evaluated based on the BJH method desorption isotherm branch, and the obtained values were included in Table 1. Considering the information obtained from the adsorption-desorption isotherms and the pores characteristics corresponding to the sampled BJH isotherms, it can be claimed that this is a material with small pores in terms of diameter and volume.

The sample’s specific surface area exhibits a significant increase compared with FA, which provides more adsorption sites used for cadmium, copper, and lead cations removal as well as a sufficiently large pore volume required for heavy metal cation trapping. The ZFAGPV-S sample shows an eight-times higher specific surface, and the ZFAGPV-A sample show a 10x higher specific surface compared with the FA.

Energy dispersive X-ray spectroscopy (EDX) was used to evaluate the surface elemental composition, which considers the elements’ energy values found in the samples. However, the EDX only provides limited quantification based on the beam penetration index and surface coverage. The results are presented in element Wt % for precursors (FA, PV glass) and ZFAGPV-A and ZFAGPV-S surfaces before and after loading with Cd^2+^, Cu^2+^, and Pb^2+^ cations (Table 2). The analyses were completed after the sample was rinsed with pure water and dried at 105 °C for 5 h.

The morphological (SEM and AFM) and elemental (EDX) evaluations of the zeolite materials (ZFAGPV-A, ZFAGPV-S) show a relatively uniform and ordered nanoparticle dispersion. The presence of active energy sites indicates a favorable situation for adsorption application in heavy metals removal. This assumption is supported by other studies on FA surface modification, indicating the influence of hydrothermal processes or alkali treatment [23,24]. In the first steps, the adsorption occurs inside the larger micropores, and the distribution of the curve is moved to the smaller pores’ diameter values (see Figure 3), confirming the roughness values.

#### 3.2.2. The Surface Energy

An OCA-20 contact angle meter from Data Physics Instruments was used to evaluate the static angle by employing a sessile drop technique. The polar and dispersive components of the surface energy play an essential role in the surface wetting behavior and, consequently, in the pollutant adsorption by the zeolite substrate. An experimental study showed that the areas with roughness lower than 0.1 µm quickly change the water drop size [51]. The zeolite materials’ contact angle evaluated using water as the interface liquid recorded low values (17.8–18.70° and 10.4–11.60°), proving a good surface wettability (Table 3), which represents an advantage in adsorption. The polar and dispersive components of the surface energy (σ) were evaluated by using two substances with much lower polarity: glycerol (with σ_s_^p^ = 41.50 mN/m polar component) and water (with 72.45 mN/m polar component). Based on the estimation developed by Owens and Wendt, 1969 [52], the ZFAGPV-A, ZFAGPV-S substrates’ surface energies exhibit a majority polar component, which is characteristic of a hydrophilic surface with good wettability and is favorable for adsorption application.

The ZFAGPV-A and ZFAGPV-S substrates with a large polar component of the surface energy (σ_s_^p^ > σ_s_^d^) and a high crystalline can better absorb heavy metal ions.

#### 3.2.3. FT-IR Evaluation

The FT-IR spectra (Figure 4) of the zeolites samples ZFAGPV-S and ZFAGPV-A before the adsorption of heavy metals were conducted to evaluate the functional group vibration frequency modifications in the zeolite, which consist of the number of adsorption bonds corresponding to chemical bonds (functional groups), proving the adsorbent elaborating nature, as shown in Table 4.

The peak intensities were recorded at 968 and 1062 cm^−1^, which consist of an asymmetric stretch of Si–Al–O; Al–O; Si–O. The adsorption band around 416 and 436 cm^−1^ corresponds to an O–T–O bending vibration, 672 and 677 cm^−1^ to a T–O–T (T = Si, Al) symmetric stretching vibration, and 968 cm^−1^ to an asymmetric stretching vibration representing the formation of Si–O–Al bonds in a sodalite and analcime structure [28,37]. The adsorption bands corresponding to OH group stretching vibrations are located at 3044 cm^−1^ (ZFAGPV-A) and 3682 cm^−1^ (ZFAGPV-S). At 2065–2357 cm^−1^, a tight and small band was present in both samples, which was attributed to CO_2_ molecule antisymmetric stretching vibrations.

### 3.3. Adsorption Experiments

#### 3.3.1. Adsorption Efficiency Function of the Contact Time in Monocation Solutions

Heavy metals adsorption tests were performed by applying an adsorbent dosage (ZFAGPV-S/ZFAGPV-A:monocation solution of 0.1 g:50 mL) at ambient temperature (22 ± 1 °C) under magnetic stirring (100 rpm) using Cd^2+^, Cu^2+^, and Pb^2+^ as cation models prepared from synthetic systems based on cadmium chloride (CdCl_2_·2.5H_2_O), copper chloride (CuCl_2_·2H_2_O), and lead chloride (PbCl_2_·2H_2_O) purchased from Scharlau Chemie S.A. with 98% purity. The second adsorption experiments were based on the simultaneous removal of three cations with concentrations of Cd^2+^ = 0/…/500 mgL^−1^, Pb^2+^ = 0/…/1000 mgL^−1^, and Cu^2+^ = 0/…/300 mgL^−1^. Aliquots were recorded at various periods until reaching a maximum of 180 min. The substrate was removed by employing a 0.45 μm filter, then the supernatant was investigated using an atomic adsorption spectrometer from Analytic Jena (Model ZEEnit 700) at the corresponding cations wavelength: λ_Cd_= 228.8 nm, λ_Pb_ = 283.30 nm, λ_Cu_= 324.75 nm, after calibration.

The removal efficiency of the pollutants [%] and the amount of heavy metal uptake *q* (mg/g) by the adsorbents (adsorption capacity) was evaluated considering the initial concentration $c_{HM}^{i}$ (mg/L), equilibrium concentration $c_{HM}^{e}$ (mg/L) of the pollutants, *V_sol_* (L) solution volume, and *m_ss_* (g) of the adsorbent using Equations (2) and (3).
(2)$$\eta \left(\%\right)=\frac{\left(c_{HM}^{i}-c_{HM}^{e}\right)}{c_{HM}^{i}}\times 100$$
(3)$$q=\frac{\left(c_{HM}^{i}-c_{HM}^{e}\right)\times V_{sol}}{m_{ss}}$$

The results of the Cd^2+^, Cu^2+^, and Pb^2+^ cation removal from the solutions on the substrates are presented in Figure 5.

The results were used to make a comparative adsorbents evaluation and tested in similar conditions considering the effectiveness of the treatment parameters applied on each substrate for different ionic species. The removal percentages of the Cd, Cu, and Pb cations from the monocation solutions applied on the ZFAGPV-A- and ZFAGPV-S-synthesized adsorbents were Pb^2+^ > Cu^2+^ > Cd^2+^. It was observed that the removal rates of the studied cations boost when the contact time increases. The optimal time for all adsorbent substrates is about 90 min, a technologically feasible time. It was found that the adsorption process has two steps (Figure 5). Due to a large number of available surface-active sites, in the first step, it fast and intensive adsorption was observed, which happen in the first 10 min. In the second step, after the first 10–20 min, the adsorption process was slower, indicating a close proximity with the equilibrium state. At the optimal contact time, the maximum adsorption percentages of lead, copper, and cadmium from ZFAGPV-A were 87.63%, 87.304%, and 67.73%, and the maximum adsorption percentages of lead, copper, and cadmium from ZFAGPV-S were 86.63%, 66.36%, and 62.37%. 

The results show that the ability of adsorption for zeolite materials—analcime is much better than for zeolite materials—sodalite. Analcime zeolite is a material with a large surface area and larger pores diameter, which favors cation absorption.

#### 3.3.2. Evaluation of the Removal Percentage and Contact Period in the Presence of Three-Cation Solutions

Evaluation of the adsorption capacity at the same initial concentration of 0.01 N of cadmium, copper, and lead cations from a three-component system is presented in Figure 6.

In this case, the selectivity orders prove that electronegativities, cation size, hydration number of cations (Table 5), and bond strength influence the adsorption capacity of zeolite materials. Based on the research carried out by Mehdipour Ghazi on copper (II), lead (II), and cadmium (II) cation removal from wastewater in systems containing one metal ion, the zeolites with different structural models indicate a significant affinity for ionic exchange with Pb(II) and reduced affinity for Cu(II) and then for Cd(II) [53]. One explanation is related to the ionic radius, considering that Pb(II) ions are the larger ions from the system. That is, being in proximity to the zeolite active sites, atoms may induce higher interactions with the substrate [54].

In conclusion, the affinity of both samples was bigger for lead than for copper and cadmium ions. Similarly, the results indicate the adsorption achievement of the hydrated ions from aqueous solutions, which may explain the increase in efficiencies for copper cations characterized by a lower volume compared with other hexahydrate cations. Figure 6 shows that the adsorption properties of ZFAGPV-A and ZFAGPV-S samples for Pb^2+^ and Cu^2+^ cations are better than those of Cd^2+^ cations.

The favorable adsorbent is ZFAGPV-A, with a higher surface area (63.3 m^2^/g), 20.527 nm roughness, and bigger pore diameter of 19.848 nm compared to 11.671 nm corresponding to ZFAGPV-S (49.83 m^2^/g). These results may suggest that the adsorption process occurs at the inner interface of the substrate pores. Adsorption occurring in the pores’ interior is considerably lower than that taking place in the pores’ exterior due to the cation transition, which starts from the exterior surface and ends in the inner pores’ surface.

#### 3.3.3. The Correlations between the Adsorbent Mass and Adsorption Efficiency

The adsorbent quantity may play an important role in increasing heavy metal adsorption efficiency. The adsorption properties of synthetic analcime (ZFAGPV-A) and sodalite (ZFAGPV-S) were evaluated by employing different adsorbent dosage quantities, and the efficiencies are presented in Figure 7. When the adsorbent is used in larger quantities, the adsorption of cations improves because the number of active centers increases, which is favorable for adsorption of Cd^2+^, Cu^2+^, and Pb^2+^ cations. Due to the increase of the substrate mass, the lead cations were almost completely removed even when the pH increased above 7, and the tendency for precipitation formation increased [18,46]. Figure 7 shows that when the substrate dosage increased, the synthetic analcime (ZFAGPV-A) adsorption rate gradually increased more than sodalite (ZFAGPV-S). When the solid/liquid ratio was 0.5 g/50 mL, the adsorption rate could reach 98.28% for lead and copper cations and 80.12% for cadmium cations. The adsorbent adsorption sites are limited and, consequently, the adsorption efficiency can be improved by optimizing the adsorbent dosage.

#### 3.3.4. Influence of the Initial Heavy Metals Concentration

The correlations between the Cd^2+^, Cu^2+^, Pb^2+^ cation initial concentrations and the conditions of 90 min contact time, 0.5 g/50 mL dosage of adsorbents, 22 °C temperature, and pH of 7.4 are presented in Figure 8. It was observed that when the initial concentration increased, there was a similar tendency for the heavy metal adsorption capacity while the adsorption efficiency records a slight decrease. The quantity of Cd^2+^ cations adsorbed during the equilibrium stage on the ZFAGPV-A sample increased by 13,14 times when the initial concentration dosage was adjusted from 18.87 mgL^−1^ to 554.1 mgL^−1^. The heavy metal ions removal percentage (%) was reduced from 99.44% to 44.49%. In a similar way, for Pb^2+^ cations adsorbed on ZFAGPV-A, the initial concentration dosage was adjusted from 17.54 mgL^−1^ to 1011 mgL^−1^, and the adsorption capacity increased 40.93 times while the adsorption efficiency (%) reduced from 99.31% to 53.49%. The adsorption capacity of the cations on the ZFAGPV-S substrate indicates values lower than the adsorption capacity of cations on ZFAGPV-A.

Figure 8a,b shows that when the initial concentration of heavy metals increases, the heavy metal removal from the aqueous solution will increase. This process is sustained by the gradient increase in concentration, which determines the adsorption driving force to increase as well [25]. It seems that when a solution with higher concentrations is employed, the time needed to attempt the adsorption equilibrium is significantly longer.

#### 3.3.5. Establishing the PZC Values of the Substrates

Adsorption is a surface process influenced by surface charges. The point of zero charges (PZC) is considered the value where the surface pH is neutral. The surface charge of the adsorbent is influenced by the solution pH and should be evaluated considering the PZC value(s). A positive substrate surface charge is obtained when the aqueous pH solution value is less than that of pH_PZC_. In this case, the sample adsorption properties for cadmium, copper, and lead ions decrease because of the competition between the heavy metals cations and H_3_O^+^. When the solution pH is higher than that of PZC, the sample surface is loaded with negative charges, and the adsorption capacity increases. The PZC for ZFAGPV-A and ZFAGPV-S substrates was evaluated based on the potentiometric titration method, and the results indicate two inflection points (Figure 9) that are related to the occurrence of surface multistep equilibrium processes. The previous study indicates that the working pH value inducing the highest heavy metal ions adsorption was pH = 7.0 for lead, 6.5 for copper, and 7.0 for cadmium. According to the Pourbaix diagram at pH 6.5–7.0, copper and cadmium cations can be present in aqueous solutions as Cu^2+^/Cu(OH)^+^ and Cd^2+^/Cd(OH)^+^ [54]. At pH = 7, lead is present in two ionic forms Pb^2+^/Pb(OH)^+^; thus, the precipitation of Pb^2+^ is negligible.

### 3.4. Modeling of Adsorption Kinetics

Two important physical-chemical criteria during the adsorption processes evaluation are represented by the adsorption kinetics and the adsorption equilibrium. The adsorption mechanism involves multiple steps: (i) the heavy metals cation transition starts from the bulk solution toward the adsorbent surface, (ii) transfer based on diffusion over the boundary layer reaches the adsorbent surface, and (iii) on the last step, there is a diffusion that takes place between the particles found into the adsorbent interior surface. Figure 10 indicates the absorption kinetics corresponding to ZFAGPV-A and ZFAGPV-S for cadmium, copper, and lead ions removal.

The adsorption kinetics experimental data were evaluated using pseudo-first- and pseudo-second-order as well as the intraparticle diffusion template required to obtain kinetic information, such as the uptake rate of the adsorbent, adsorption capacity, and kinetic mechanism, respectively. 

The kinetics based on the pseudo-first-order model uses the following equation [56]: (4)$$\log\left(q_{e}-q_{t}\right)=\log\left(q_{e}\right)-\frac{K_{L}}{2.303}t$$
where *K_L_* (min^−1^) is the Lagergren constant calculated from log(*q_e_* − *q_t_*) plot over *t*, the adsorbed pollutant quantities (mg/g) at equilibrium are represented by *q_t_* and *q_e_*, and *t* (min) represents the time required to reach the equilibrium stage. 

The kinetics model corresponding to pseudo-second-order Equation (5) was considered for monolayer adsorption analysis [57,58]:(5)$$\frac{t}{q_{t}}=\frac{1}{k_{2}q_{e}^{2}}+\frac{t}{q_{e}}$$
(6)$$h=k_{e}q_{e}^{2}$$
where the pseudo-second-order adsorption rate constant (*k*_2_) was evaluated based on *t*/*q_t_* versus *t* plotting. *h* represents the initial adsorption rate (mg g^−1^min^−1^) and was evaluated using Equation (6). The main assumption attributed to this model consists of the fact that the rate-limiting step is based on a chemisorptions processes using valence forces for adsorbent-adsorbate electron transition [59]. 

The diffusion model between particles corresponds to Equation (7) [60]:(7)$$q=k_{id}t^{1/2}+C$$
where the rate constant for intraparticle diffusion is *k_id_*, and the adsorbent boundary layer thickness is *C* (mg/g). The resulting values from the kinetic evaluation can be found in Table 6.

The values of the correlation coefficients ($R_{Pb}^{2}=1$, $R_{Cu}^{2}$= 0.999, and $R_{Cd}^{2}=0.999$) and a desirable match between the experimental q_e(exp)_ and theoretical values q_e(cal)_ indicate that Pb^2+^, Cu^2+^, Cd^2+^ cation adsorption onto ZFAGPV-A and ZFAGPV-S can be perfectly expressed based on the pseudo-second-order model, proving a chemisorptions mechanism. The heavy metals adsorption on ZFAGPV-A occurs fast, with k_2_ = 0.0342 g mg^−1^min^−1^ and k_1_ = 0.0157 min^−1^ (Table 6). Consequently, the chemisorptions are indeed a rate-limiting step [61]. In the particular situation of C > 0, the rate-limiting steps are represented by the external mass transfer and diffusion between particles [62]. The adsorption occurring on ZFAGPV-A and ZFAGPV-S is considered a multistep process including both internal pores diffusion and external surface adsorption. The copper, cadmium, and lead adsorption from the three-cations solution on the zeolite substrate is well fit with the second-order kinetic model, where R^2^ = 0.968 vs. q_e(Cd)_ =181.81 mg/g, R^2^ = 0.993 vs. q_e(Cu)_=111.11 mg/g, and R^2^= 999 vs. q_e(Pb)_ = 400.00 mg/g. The results indicate that the metal ions-adsorbent surface interaction consists of a monosite adsorption.

### 3.5. Adsorption Batch Isotherms

The evaluation based on the adsorption isotherms provides information regarding how the adsorbed cations are dispersed in two states of matter (liquid and solid phase) when equilibrium is reached [63]. The metal cations adsorption isotherms are shown in Figure 11 for the ZFAGPV-A and ZFAGPV-S samples. In accordance with the correlation coefficient R^2^ for both samples and all metal cations, the Langmuir isotherm is most suitable to describe the adsorption process.

The adsorption isotherms are based on the Langmuir and Freundich models.

The Langmuir isotherm is expressed by the linearized equation, Equation (8):(8)$$\frac{c_{e}}{q_{e}}=\frac{1}{q_{max}K_{a}}+\frac{c_{e}}{q_{max}}$$
where the monolayer maximum adsorption (mg·g^−1^) capacity is *q_max_*; the adsorption constant is *K_a_* (L·mg^−1^); the overall adsorption capacity at equilibrium is *q_e_*; the heavy metals ions concentration (mg·L^−1^) at equilibrium is *c_e_*. The *K_a_* and *q_max_* values were calculated based on the slope-intercept corresponding to the *c_e_*/*q_e_* vs. *c_e_* plot. The value of R_L_ representing the separation factor is evaluated using Equation (9).
(9)$$R_{L}=\frac{1}{1+K_{a}C_{0}}$$

*R_L_* characterizes favorable adsorption if smaller than 1 [64,65].

The Freundlich empirical isotherm was employed to evaluate the adsorption occurring in dilute solutions. The regular adsorption isotherm is represented by the following equation:(10)$$q_{e}=K_{F}C_{e}^{1/n}$$

The Freundlich isotherm is expressed by the linearized equation, Equation (11):(11)$$lnq_{e}=lnK_{F}+\frac{1}{n}C_{e}$$
where the adsorption capacity corresponding to the Freundlich constant is represented by *K_F_*, and the adsorption density as a dimensionless parameter is represented by 1/*n*. Table 7 contains the parameters obtained from the two isotherms.

The heavy metal ions adsorption originating from the aqueous solution on the surface substrate samples is well described by the Langmuir isotherm model. This isotherm can be used to describe monolayer surface adsorption showing a limited number of identical active sites. The kinetic model considers that there is no adsorbate-surface transmigration, and the adsorption energies are uniform for all sites [66]. There is a correlation between the Langmuir constant (*K_a_*) and adsorption energy of adsorption, showing that the quantity of adsorbed heavy metals increases when the adsorption energy is lower. The analogic evaluation of the results for both samples, ZFAGPV-A and ZFAGPV-S, indicates that the maximum monolayer coverage capacity is higher for ZFAGPV-A: 62.112 mg/g (Cd^2+^), 55.556 mg/g (Cu^2+^), and 175.439 mg/g (Pb^2+^), with lower values of (33.445 mg/g (Cd^2+^), 54.945 mg/g (Cu^2+^) and 158.731 mg/g (Pb^2+^) for ZFAGPV-S, respectively. The Langmuir model well describes the samples’ adsorption behavior, exhibiting correlation coefficients (R^2^) between 0.951–0.999. According to the q_max_ parameter, zeolitic materials adsorption occurs in the following order: Pb^2+^> Cu^2+^> Cd^2+^. The literature indicates that nanocomposites based on a magnetic iron oxide-silica shell exhibit a Pb^2+^ maximum adsorption capacity of 14.9 mg/g [67] while waste-toned power materials show a 3.49 mg/g Cd^2+^ adsorption capacity [68].

The theoretical adsorption capacity corresponding to the tested substrates is similar to the experimentally obtained values. It must be underlined that for 0 < R_L_ <1, the results are close to 1, which shows a liner adsorption characteristic. 

The K_F_ constant is used as an adsorption capacity approximate indicator in the Freundlich model. Even if the experimental data exhibit low correlation coefficients when the Freundlich model is employed, in analogy with the Langmuir model, a similar adsorption capacity hierarchy was observed: ZFAGPV-A > ZFAGPV-S. When the 1/n is below 1, the process is considered a normal Langmuir isotherm. Contrary, when 1/n is higher than 1, then the process is a cooperative adsorptions process [64,66]. Additionally, for all experimental evaluations, the 1/n ratio provides values between 0.305–0.486, corresponding to the normal model of the Langmuir isotherm [69,70]. On both substrates, the Langmuir model describes a heavy metals cations chemisorption mechanism based on the premise that the monolayer adsorption formed on the substrate is provided by the electrostatic attractions. The cations adsorption that occurs is supported by a monolayer mechanism; however, the presence of other adsorptions types cannot be excluded. Over the course of the adsorption mechanism, there is a heavy metal cations transition through the pores and lattice channels in order to switch with exchangeable Na^+^ and K^+^ cations [71]. The metal cations transfer from the top layers of the intercrystalline areas is viewed as a speed control step in the ion exchange [72]. The zeolite materials ZFAPVG-A and ZFAPVG-S contain alkali cations Na^+^ and K^+^, so the ion-exchange process happens between alkali cations and Cu^2+^, Cd^2+^, and Pb^2+^ cations during contact between the cations and zeolite particles (Equation (12)). The adsorption process takes place with good efficiency because on the substrate surface, several active centers, such as (≡SiO–) and (≡AlO–), were formed. In the presence of heavy metal (HM) cations, these centers form complex structures, such as (≡Si–O)_2_HM and (≡Al–O)_2_HM, [70].
(12)$$Zeolite\;\equiv n\left(Na^{+},\;K^{+}\right)+M^{n+}\to Zeolite\;\equiv \;M^{n+}+n\left(Na^{+},\;K^{+}\right)$$

*M*^*n*+^ can be *Cu*^2+^, *Cd*^2+^, and *Pb*^2+^ cations.

The zeolites’ intrinsic characteristics suggest these materials for different applications, such as water purification, membrane separation, absorbability, antimicrobial activities, and coagulation processes. The main advantage of zeolites consists of their structural diversity and porosity [73,74]. Different zeolites exhibit catalytic properties due to the Bronsted acid active sites on OH groups bridging the framework between the aluminum and silicon channel. Nowadays, zeolite materials can be used in wastewater treatment, on large-scale industrial applications, or even in hard water treatment. The high absorption induced by their porosity can be used not only for heavy metal removal but also for eliminating other impurities, phosphorus, ammonium, etc. [75].

## 4. Conclusions

Raw coal fly ash and PV glass were modified to zeolites ZFAGPV-A and ZFAGPV-S using fusion and hydrothermal methods. The effectiveness and efficiencies of the zeolites ZFAGPV-A and ZFAGPV-S, adsorbents substrates developed from PV glass and coal fly ash for Cd^2+^, Cu^2+^, and Pb^2+^ cations adsorption from wastewater, were studied in this work.

It was found that zeolite with analcime (ZFAGPV-A) and zeolite with sodalite (ZFAGPV-S) show superior removal efficiencies compared to raw coal fly ash (FA). The highest equilibrium adsorption capacity was established during the first 90 min of immersion when the adsorbent dose was 10 g/L. The data show that the cations’ initial dosage has a significant influence on the adsorption sample characteristics of ZFAGPV-A and ZFAGPV-S, proving that when the initial cation dosage is higher, the zeolite adsorption capacity increases. The kinetic studies indicate a good correlation (R^2^ > 0.9999) of the experimental results with a pseudo-second-order model.

Heavy metal ions adsorption from aqueous solutions is based on the electrostatic forces and depends on the polar component of the zeolite surface energy materials (ZFAGPV-A, ZFAGPV-S), which can strongly influence the adsorption capacity. The adsorption of cadmium, copper, and lead cations on zeolites ZFAGPV-A and ZFAGPV-S within these initial concentrations range can be explained using Langmuir and Freundlich models. Both constant values, n from the Freundlich constant and the separation factor R_L_ from the Langmuir model, provided a suggestion for favorable adsorption. ZFAGPV-A has the highest adsorption capacities, q_max_, and could be prioritized in units of mg/g: Pb^2+^ (175.439) > Cu^2+^ (60.112) > Cd^2+^ (55.556) at pH after pH_PZC_.

The highest adsorption of Cu^2+^ vs. Cd^2+^ may be related to the fact that the radius of copper cations is smaller than cadmium cations. These new zeolite materials developed from two wastes, coal fly ash and PV glass, can contribute to the purification of soil, water, and wastewater.

## Data Availability

The data presented in this study are available by requesting from the corresponding author.

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
