# Peer review of "Opportunities for Recycling PV Glass and Coal Fly Ash into Zeolite Materials Used for Removal of Heavy Metals (Cd, Cu, Pb) from Wastewater"

_materials, 2022, doi:10.3390/ma16010239_

Round 1

Author Response

Dear Reviewer,

We express our gratitude to your work and guidance that was helping us to improve the quality of this manuscript.

We have considered all your comments and suggestion in the new revised form of the manuscript. The changes where highlighted in red.

Q1. The cited articles 4 and 5 are the same. The cited articles 34 and 41 there is an error in the citation titles. Please correct that mistake.

A1. Thank you for the observations. We made the necessary corrections:

Ref. 4: “Duta, A.; Andronic, A. and Enesca, A. The influence of low irradiance and electrolytes on the mineralization efficiency of organic pollutants using the Vis-active photocatalytic tandem CuInS2/TiO2/SnO2. Catal. Today 300 (2017) 18-27.”

Ref. 5: “H. Patel, R.T., Vashi, Treatment of textile wastewater by adsorption and coagulation. E- Journal of Chemistry. 7,(2010) 1468–1476.”

Ref. 34: “Y. Li, L. Li, J. Yu, Applications of zeolites in sustainable chemistry, Review. Chem. 3, (2017) 928-949.”

Ref. 41: “A. Molina, C. Polle, A comparative study using two methods to produce zeolites from fly ash. Miner. Eng. 17 (2004) 167–173.”

Q2. The text in section "Zeolite materials synthesis and characterization" You get zeolite by combination of alkaline fusion and hydrothermal treatment of the fly ash and PV glass with different amounts of NaOH are analcime, sodalite, chabazite K, ZSM-5(H), zeolite N, cancrinite and tobermorite. Then it is not clear ZFAGPV-S and ZFAGPV-A which exact type of zeolite is obtained.

A2. Based on your comment we have improved the explanations regarding the materials used in this manuscript.

Lines:

Q3. The XRD analysis the resulting zeolites are not a pure phase.

A3. The reviewer in right. We have underlined this aspect in the manuscript.

Lines:

Q4. The BET analysis from the presented results, the obtained materials have a very low surface area, while in the article they wrote "The huge specific surface area provided a large number of adsorption sites for the adsorption of cadmium, cooper and lead cations on the surface of the substrates." Which in my opinion the interpretation of the results does not correspond with the results of the table.

A4. Indeed, the SBET is not huge. We made this observation by comparing the FA SBET with the SBET of the final sample. ZFAGPV-S sample has the SBET 8x time higher than that of FA and ZFAGPV-A sample has the SBET 10x time higher than that of FA. We have modified the sentence in order to be clearer.

Lines:

Q5. The FTIR spectra characteristic bands at 3619, 3645, and 3692 cm-1 represent OH groups stretching vibrations are indicated in the text, while these bands are missing in the graph. Please correct that mistake.

A5. We have corrected the mistake. Thank you for the observation.

Lines:

Q6. On page 12, the text describes surface area and pores diameter, which have no correspondence in table 1. Please correct that mistake.

A6. Thank you for the comment. We have corrected the mistake.

Lines:

Q7. Are not shown graphs of adsorption isotherms, equilibrium isotherms and adsorption kinetics. Аdsorption kinetics and adsorption equilibrium by using Langmuir and Freundlich models, should be considered for at least three concentrations.

A8. We have inserted the graphs corresponding to the adsorption kinetics as suggested. We agree that multiple concentrations can be used for surface evaluations. However, in this study we made the incipient evaluation of the heavy metal removal. A dedicated paper based on adsorption studies will be prepared in the near future.

Reviewer 2 Report

The authors have studied the formation and applications of zeolite materials from PV glass and fly ash, and presented detailed characterization and experiment results. This method can be one possible approach to recycle the above solid waste, however, the actual benefit of utilizing the PV glass and fly ash is not clearly discussed. Several issues are to be addressed:

1)     The equipment for zeolite synthesis was not described.

2)     What is the insert in Figure 2(a)? It is not discussed.

3)     Line 184, it is not appropriate to get the particle size from SEM picture; besides, how do the authors determine the particle is analcime. The values in the picture are not clear either.

4)     AFM characterization seems not meaningful to the reviewer, as the roughness depends not only on the particles, but also the way the AFM samples are prepared and the dispersion of particles. Please provide an argument.

5)     Table 2, the EDX data shows 30.99% of Pb element adsorbed on ZFAPVG-A, which is quite high. Was the sample rinsed before drying?

6)     In section 3.3.1, the mono cation concentration is 2000ppm, how realistic is this concentration for heavy metal? The real wastewater contains much lower heavy metal concentration.

7)     Line 280, what is the meaning of ÷ 120 ÷ 180?

8)     Figure 6, why was the adsorption of copper higher than lead at the beginning?

9)     What is the key parameter determining the adsorption efficiency? In line 337-338, the roughness and pore diameter may not play an important role.

10)  Section 3.3.3 is trivial fact. No new findings are presented.

11)  Line 396-388, is the lower adsorption capacity mostly due to the repelling of like charges, or the competition of proton?

12)  Figure 9 shows two values of pHPZC, which is the correct one?

13)  Table 6 is slightly distorted, please align.

14)  There are numerous grammar and typo errors in the manuscript, it would be very helpful if the authors can get some professional editing service.

Author Response

Dear Reviewer,

We express our gratitude to your work and guidance that was helping us to improve the quality of this manuscript.

We have considered all your comments and suggestion in the new revised form of the manuscript. The changes where highlighted in red.

Q1. The equipment for zeolite synthesis was not described.

A1. We have inserted more details related with the equipment used for zeolite synthesis.

Lines:

Q2. What is the insert in Figure 2(a)? It is not discussed.

A2. The insets from Figure 2a represents the SEM images of the raw materials used to obtain the zeolites samples. We have inserted an additional explanation to be clearer.

Lines:

Q3.  Line 184, it is not appropriate to get the particle size from SEM picture; besides, how do the authors determine the particle is analcime. The values in the picture are not clear either.

A3. We have inserted visible values in the picture. We agree that SEM can’t give an overall evaluation of the particle size. This is why we have highlighted this aspect into the manuscript.

Lines:

Q4.  AFM characterization seems not meaningful to the reviewer, as the roughness depends not only on the particles, but also the way the AFM samples are prepared and the dispersion of particles. Please provide an argument.

A4. Indeed, the sample preparation and analysis procedures may influence the AFM results. We will like to outline that the analyses were made on several points to ensure the results representativity. We have included supplementary explanations into the manuscript.

Lines:

Q5. Table 2, the EDX data shows 30.99% of Pb element adsorbed on ZFAPVG-A, which is quite high. Was the sample rinsed before drying?

A5. Thank you for the comment. We have included into the manuscript the steps made prior the EDX analysis which include rinsing and drying procedures.

Lines:

Q6. In section 3.3.1, the mono cation concentration is 2000ppm, how realistic is this concentration for heavy metal? The real wastewater contains much lower heavy metal concentration.

A6. High concentrations of heavy metal can be found in the galvanization industrial activity as well as some mining operations. We have considered an average value between these components which are responsible for the main discharges in water surfaces.

Q7.   Line 280, what is the meaning of ÷ 120 ÷ 180?

A7. There was a mistake in editing. We made the necessary corrections.

Lines:

Q8. Figure 6, why was the adsorption of copper higher than lead at the beginning?

A8. This behavior was observed by other authors as well [1-4]. It is supposed to be influenced two main factors: the ionic radius which is bigger for lead compared copper and cadmium and the interface-electronegativity dependent process (lead has the highest electronegativity).

[1] M. Medykowska, M. Wiśniewska, R. Panek, Interaction mechanism of heavy metal ions with the nanostructured zeolites surface – Adsorption, electrokinetic and XPS studies. Journal of Molecular Liquids, Volume 357, 1 July 2022, 119144.

[2] Y. Lv, B. Ma, Y. Chen, Adsorption behavior and mechanism of mixed heavy metal ions by zeolite adsorbent prepared from lithium leach residue. Microporous and Mesoporous Materials, Volume 329, January 2022, 111553.

[3] Z. Mo, D.Z. Tai, A. Shahab. A comprehensive review on the adsorption of heavy metals by zeolite imidazole framework (ZIF-8) based nanocomposite in water. Chemical Engineering Journal, Volume 443, 1 September 2022, 136320.

[4] M.F. Mubarak, A.M.G. Mohamed, N. Shehata, Adsorption of heavy metals and hardness ions from groundwater onto modified zeolite: Batch and column studies. Alexandria Engineering Journal, Volume 61, Issue 6, June 2022, Pages 4189-4207.

Q9. What is the key parameter determining the adsorption efficiency? In line 337-338, the roughness and pore diameter may not play an important role.

A9. We have improved to sentence in order to be clearer that the adsorption efficiency depends on the specific surface area and pores diameters.

Lines:

Q10.  Section 3.3.3 is trivial fact. No new findings are presented.

A10. We understand your opinion. However, there are reviewers who made the statement that the articles address to a large spectrum of people who may be interested by the aspects presented in this section. More exactly, they may be interested on the magnitude of the efficiency variations based on the substrate dosage as well as the moment when the adding more substrate will not significantly change the removal efficiency.

Q11. Line 396-388, is the lower adsorption capacity mostly due to the repelling of like charges, or the competition of proton?

A11. We consider that the repelling of the like charge may cause the decrease of adsorption capacity. We are still making several experiments for a future paper in order to explain in more details this behavior.

Q12.  Figure 9 shows two values of pHPZC, which is the correct one?

A12. Both values are correct. We found two inflection points that can be associated with the multiple equilibrium processes produced at/on adsorbent surface.

Q13. Table 6 is slightly distorted, please align.

A13. Thank you for the observation. The final alignment will be done on the corrected proof as the template parameters change during the conversion. We are in contact with the editing staff for these aspects.

Q14. There are numerous grammar and typo errors in the manuscript, it would be very helpful if the authors can get some professional editing service.

A14. Thank you for the suggestion. We will proceed with your advice during the proof correction.

Reviewer 3 Report

This is an interesting and well written study reporting the metal adsorption on two kinds of sorbents, obtained from recycling industrial silicon containing materials. The paper can be recommended for publications after some improvements as listed below.

Major remarks

1. Figure 8 is very hard to read and interpret. In addition, it is not explained, what is the difference between plots a and b.

Please replace the figure and provide x-y type plots for all data.

2. Please provide the metal ion adsorption isotherm plots, together with the fitted isotherm model functions, preferably in their origina, non-linear form.

3. The performance and application suitability of the obtained materials can be understood, when compared to other, similar, or related sorbent materials. Please provide a comparison with other silica-based sorbent materials.
Among others, these recent studies of the same metal ions can be mentioned for comparison:

https://doi.org/10.3390/app10082726
https://doi.org/10.3390/gels8070443
https://doi.org/10.3390/ma15124150

Minor remarks

Table 6: the wt% values do not add up to 100 in the last column. Please check and complete the table.

Many grammar errors can be found in the text. Please check carefully the manuscript. Please change cooper to copper throughout the text.

These are some of the items to be corrected:

line 126 check grammar

line 129 use the proper symbol for theta

line 163 I am not sure, what is the meaning of "syn"

line 201 check grammar

line 211 check grammar

line 259 please check the phrase " sharp and intensity"

line 331 please revise and correct the sentence.

line 472-473 should be: comparing the data

line 517-519 please rewrite this sentence, to state clearly what happened and what is the conclusion.

Refs. 14, 22,32,39,62 please correct the abbreviated title of the journals, and check all other abbreviations as well.

Author Response

Dear Reviewer,

We express our gratitude to your work and guidance that was helping us to improve the quality of this manuscript.

We have considered all your comments and suggestion in the new revised form of the manuscript. The changes where highlighted in red.

We wish to acknowledge the reviewer for his patience to outline the grammar mistakes.

Q1. Figure 8 is very hard to read and interpret. In addition, it is not explained, what is the difference between plots a and b.

Please replace the figure and provide x-y type plots for all data.

A1. We have improved the quality representation of Figure 8. The difference between plots a and b is presented in the figure caption. The 3D representation is able to integrate all the necessary information’s. The axes plots and significance are now clearly presented.

Q2. Please provide the metal ion adsorption isotherm plots, together with the fitted isotherm model functions, preferably in their origina, non-linear form.

. We have included in the revised manuscript new figures and correlations related with the adsorption isotherm plots. Figure 10 and the subsequent comment are based on the cadmium, cooper and lead isotherms corresponding to ZFAGPV-A and ZFAGPV-S samples.

Q3. The performance and application suitability of the obtained materials can be understood, when compared to other, similar, or related sorbent materials. Please provide a comparison with other silica-based sorbent materials.
Among others, these recent studies of the same metal ions can be mentioned for comparison:

https://doi.org/10.3390/app10082726
https://doi.org/10.3390/gels8070443
https://doi.org/10.3390/ma15124150

A3. Thank you for the recommendation. We have included two of the three references in our manuscript to compare the materials performances in metal ion adsorption.

Q4. Table 6: the wt% values do not add up to 100 in the last column. Please check and complete the table.

A4. We believe it is about Table 2. We have checked again all values and some corrections were necessary. However, the sum will not be 100% for all samples. The differences consist on the EDX device technological limitation to identify all elements present on the surface. We have underlined these aspects into the manuscript.

Q5. Many grammar errors can be found in the text. Please check carefully the manuscript. Please change cooper to copper throughout the text.

A5. Thank you for the observations. The manuscript was revised again and will be revised second time during the proof revision.

Q6. line 126 check grammar

A6. Corrected.

Q7. line 129 use the proper symbol for theta

A7. Corrected.

Q8. line 163 I am not sure, what is the meaning of "syn"

A8. We have explained the abbreviation.

Q9. line 201 check grammar

A9. We corrected the mistake.

Q10. line 211 check grammar

A10. We have changed the entire phrase.

Q11. line 259 please check the phrase " sharp and intensity"

A11. We have corrected the mistake.

Q12. line 331 please revise and correct the sentence.

A12. We have corrected the mistake.

Q13. line 472-473 should be: comparing the data

A13. We have corrected the mistake.

Q14. line 517-519 please rewrite this sentence, to state clearly what happened and what is the conclusion.

A14. We have rewritten the sentence to be clearer.

Q15. Refs. 14, 22,32,39,62 please correct the abbreviated title of the journals, and check all other abbreviations as well.

A15. Thank you for the observation. All the reference editing will be done during the proof corrections. There are other typo and table design to be arranged as well during the proof corrections.

Reviewer 4 Report

Manuscript ID: materials-2030204

Title: Opportunities for recycling the PV glass and fly ash into zeolites materials used for removal of heavy metals from aqueous solution

Authors: Maria Visa and Alexandru Enesca.

The title can be improved: Opportunities for recycling the PV glass and coal fly ash into zeolites materials used for removal of heavy metals (Cd, Cu, Pb) from wastewater.

Introduction.

Line 25-33. Authors should add references for this information.

Line 36-37. Add references for Cu content in drinking water and about Pb pollution effect.

Authors should use “coal fly ash – (CFA)” in all article text.

Line 121-123. Why need this information in this section? Authors should remove it or add to Introduction section.

Section 3.1. Authors must add the chemical composition (wt. %) of raw CFA and PV glass by XRF or ICP methods. Please don’t use the SEM-EDS method for this analysis.

Figure 1. All peaks must be sign all CFA XRD patterns. Use symbols, not chemical formulas.

Section 3.1. Authors must add the chemical composition (wt. %) of MFAPVG and ZFAPVG-A, ZFAPVG-S by XRF or ICP methods. Please don’t use the SEM-EDS method for this analysis.

Section 3.1. Authors can add the chemical reactions between CFA and PV glass during fusion and hydrothermal treatment.

Line 166. I believe that magnetite – Fe3O4, not Fe2O3. I cannot find the peak of Fe-oxide on XRD patterns on Figure 1b.

Table 1. How was changed the mean particle size of raw CFA and PV glass after fusion and hydrothermal treatment? Authors should add the LD analysis.

Table 1. The pore diameter of ZFAGPV-A must be less than ZFAGPV-S?

Table 2. Where are the SEM images of EDX analysis? Please add them.

Figure 5-6. Authors must improve these figures. The columns on 10 and 20 min overlap each other, to fix this it is necessary to use points 1-9 (not 10, 20, 30 ... min) on the X axis, then the distance will be the same, then delete the values on X axis, and sign the points in min. Then the Figure will correctly show the columns.

Table 5. Add reference for this information on table title.

Figure 7. Don't use the 0 point on the x-axis (use 0.5-0.65 scale for X axis) and remove the borders on the figure that are left from the table shape.

Figure 8. Unable to understand column values.

Article must have a discussion section. Ending the article with a reaction is not good. Make a discussion about the possible applications of zeolites with links to recent research, 1-2 paragraphs will be enough.

Author Response

Dear Reviewer,

We express our gratitude to your work and guidance that was helping us to improve the quality of this manuscript.

We have considered all your comments and suggestion in the new revised form of the manuscript. The changes where highlighted in red.

Q1. The title can be improved: Opportunities for recycling the PV glass and coal fly ash into zeolites materials used for removal of heavy metals (Cd, Cu, Pb) from wastewater.

A1. We have changed the title in order to be clearer.

Q2. Line 25-33. Authors should add references for this information.

A2. New references were added to support the information’s.

[1] Liang, J.; Yuan, Y.; Zhang, Z.; You, S.; Yuan, Y. Modeling a Three-Stage Biological Trickling Filter Based on the A2O Process for Sewage Treatment. Water 2021, 13, 1152.

[2] Jin, X.; Che, R.; Yang, J.; Liu, Y.; Chen, X.; Jiang, Y.; Liang, J.; Chen, S.; Su, H. Activated Carbon and Carbon Quantum Dots/Titanium Dioxide Composite Based on Waste Rice Noodles: Simultaneous Synthesis and Application in Water Pollution Control. Nanomaterials 2022, 12, 472.

[3] Yang, T.; Chen, M.; Wu, S. Removal Effect of Basic Oxygen Furnace Slag Porous Asphalt Concrete on Copper and Zinc in Road Runoff. Materials 2021, 14, 5327.

Q3. Line 36-37. Add references for Cu content in drinking water and about Pb pollution effect.

A3. New references were added regarding Cu concentration in drinking water and Pb pollution effect.

[4] Karim, K.; Guha, S.; Beni, R. Comparative Analysis of Water Quality Disparities in the United States in Relation to Heavy Metals and Biological Contaminants. Water 2020, 12, 967.

[5] Molaudzi, N.R.; Ambushe, A.A. Sugarcane Bagasse and Orange Peels as Low-Cost Biosorbents for the Removal of Lead Ions from Contaminated Water Samples. Water 2022, 14, 3395.

Q4. Authors should use “coal fly ash – (CFA)” in all article text.

A4. We made the change and replaced fly ash with coal fly ash.

Q5. Line 121-123. Why need this information in this section? Authors should remove it or add to Introduction section.

A5. We consider that the sentence should remain in the dedicated section. The readers can compare our work with other papers dealing with same type of materials.

Q6. Section 3.1. Authors must add the chemical composition (wt. %) of raw CFA and PV glass by XRF or ICP methods. Please don’t use the SEM-EDS method for this analysis.

A6. Thank you for the suggestion. Unfortunately, our research infrastructure doesn’t have the possibility to provide XRF or ICP analysis. This is the reason of using SEM-EDS. We understand the SEM-EDS limitations for this kind of evaluation and we have underlined into the manuscript this aspect.

Q7. Figure 1. All peaks must be sign all CFA XRD patterns. Use symbols, not chemical formulas.

A7. We made the changes based on your recommendation’s.

Q8. Section 3.1. Authors must add the chemical composition (wt. %) of MFAPVG and ZFAPVG-A, ZFAPVG-S by XRF or ICP methods. Please don’t use the SEM-EDS method for this analysis.

A8. Thank you for the suggestion. Unfortunately, our research infrastructure doesn’t have the possibility to provide XRF or ICP analysis. This is the reason of using SEM-EDS. We understand the SEM-EDS limitations for this kind of evaluation and we have underlined into the manuscript this aspect.

Q9 Section 3.1. Authors can add the chemical reactions between CFA and PV glass during fusion and hydrothermal treatment.

A9. In this manuscript we didn’t evaluate the reaction mechanism from which zeolites are formed. The paper is manly focused on the sample ability for heavy metals removal. The zeolites formation mechanism will be the subject of another paper related with synthesis parameters and their influence on the zeolites composition and structure.

Q10. Line 166. I believe that magnetite – Fe3O4, not Fe2O3. I cannot find the peak of Fe-oxide on XRD patterns on Figure 1b.

A10. We found mostly Fe2O3. However, both phases may be present. We have modified Figure 1b to outline Fe2O3 diffraction peak. The Fe2O3 in zeolites is very small.

Q11. Table 1. How was changed the mean particle size of raw CFA and PV glass after fusion and hydrothermal treatment? Authors should add the LD analysis.

A11. In the absence of LD set-up in our institute we made a statistical evaluation based on the SEM images in different points. The glass beads and the spherical particles from FA are converted in zeolites particles with different shapes and sizes. Additionally, large aggregates are formed with sizes varying from 6.3 µm up to 33.2 µm.

Q12. Table 1. The pore diameter of ZFAGPV-A must be less than ZFAGPV-S?

A12. No, the pore diameter of ZFAGPV-A is higher compared with ZFAGPV-S. The SBET is also higher in the first case.

Q13. Table 2. Where are the SEM images of EDX analysis? Please add them.

A13. The EDX analysis is not based on surface mapping but on random point by point analysis. This technique allows reducing the errors during the measurements. This is why there are no SEM corresponding images.

Q14. Figure 5-6. Authors must improve these figures. The columns on 10 and 20 min overlap each other, to fix this it is necessary to use points 1-9 (not 10, 20, 30 ... min) on the X axis, then the distance will be the same, then delete the values on X axis, and sign the points in min. Then the Figure will correctly show the columns.

A14. Thank you very much for your tips. We followed your recommendations and the figures look better.

Q15. Table 5. Add reference for this information on table title.

A15. We have added a new reference for Table 5.

Q16. Figure 7. Don't use the 0 point on the x-axis (use 0.5-0.65 scale for X axis) and remove the borders on the figure that are left from the table shape.

A16. Thank you for the suggestions. Figure 7 was improved.

Q17.Figure 8. Unable to understand column values.

A17. Figure 8 was improved and the column values are visible.

Q18. Article must have a discussion section. Ending the article with a reaction is not good. Make a discussion about the possible applications of zeolites with links to recent research, 1-2 paragraphs will be enough.

A18. We have included at the end of chapter 3 a short discussion about relevant features of zeolites and possible applications.

Round 2

Reviewer 2 Report

The authors have addressed the issues raised by the reviewer and the manuscript is suitable for publication. 

Author Response

Thank you!

Reviewer 3 Report

In the revised version, authors provided some improvement to the presentation, but unfortunately neglected some of the comments of the first review.

1.  The question - answer no.2. for the first revision:

Q2. Please provide the metal ion adsorption isotherm plots, together with the fitted isotherm model functions, preferably in their origina, non-linear form.
A2. We have included in the revised manuscript new figures and correlations related with the adsorption isotherm plots. Figure 10 and the subsequent comment are based on the cadmium, cooper and lead isotherms corresponding to ZFAGPV-A and ZFAGPV-S samples.

Present comment:

 I do not see individual isotherm plots for the metal ions, plotted together with the fitted isotherm model functions. Figure 10 is not an adsorption isotherm plot.
These are necessary data to be presented and discussed in a paper dealing with sorptive removal of metal ions. Please provide the isotherms and the fitted curves in clear, x-y graphs.

2. There are still many grammar errors in the text. Please invite a professional for a careful revision of the text.

3. There are many journal names with wrong abbreviations in the references.
 such as: 4, 8, 14, 16, 18, and possibly others.

Author Response

Dear Reviewer,

Thank you for your kind suggestions which were very helpful to improve the quality of our manuscript. We have addressed all your concerns and the answers can be found bellow.

Q1. I do not see individual isotherm plots for the metal ions, plotted together with the fitted isotherm model functions. Figure 10 is not an adsorption isotherm plot.

These are necessary data to be presented and discussed in a paper dealing with sorptive removal of metal ions. Please provide the isotherms and the fitted curves in clear, x-y graphs.

A1. Indeed, it was our misunderstanding and we apologies for this. Now we have inserted all adsorption isotherms for both samples and the three cations (Figure 11). We have added supplementary explanations to support our conclusions.

Q2. There are still many grammar errors in the text. Please invite a professional for a careful revision of the text.

A2. Thank you for the recommendation. We made another corrections and the editorial team will make the final check during the proof development.

Q3. There are many journal names with wrong abbreviations in the references.

 such as: 4, 8, 14, 16, 18, and possibly others.

A3. We made the corrections and now all the references are in MDPI style.

The authors

Reviewer 4 Report

My main questions remain the same:

Chemical analysis cannot be performed with SEM-EDX as it is a qualitative analysis and not a quantitative one. It is an important fact. If the authors do not have such equipment, this is not a reason not to do such an analysis. It is necessary to apply to other Institutions.

The same is true for laser diffraction (LD). If the authors do not have such equipment, then SEM images can be used. For statistics, it is necessary to calculate the average particle size. There should be more than 1000 measurements, then it is possible to create a figure with good reliability. That is, the authors must take many SEM images. Then, using a program (for example, Image Pro), you can correlate the pixel size regarding the scale and calculate each particle size. After calculating a thousand particles, statistics will be enough to build a figure. Authors need to do this for 3 samples. This is not a difficult task and requires 1-2 hours of time. Calculation of the average particle size should be carried out using the Sturges equation:

n = 1 + 3.22 × Log (N)

where N is the number of measured particles.

High-impact research in the Q1 journal means using standard methods to analyse samples. The “Materials” just publishes articles where a qualitative analysis of the properties of materials is carried out. If the authors want to correspond a high level, it is necessary to do the most standard methods of analysis, such as XRF, ICP and LD.

Author Response

Dear Reviewer,

We will like to send our gratitude for all your precious advices. We regret that we are not able to fulfill your exigencies in term of characterization methods and analysis numbers.  

Q1. Chemical analysis cannot be performed with SEM-EDX as it is a qualitative analysis and not a quantitative one. It is an important fact. If the authors do not have such equipment, this is not a reason not to do such an analysis. It is necessary to apply to other Institutions.

The same is true for laser diffraction (LD). If the authors do not have such equipment, then SEM images can be used. For statistics, it is necessary to calculate the average particle size. There should be more than 1000 measurements, then it is possible to create a figure with good reliability. That is, the authors must take many SEM images. Then, using a program (for example, Image Pro), you can correlate the pixel size regarding the scale and calculate each particle size. After calculating a thousand particles, statistics will be enough to build a figure. Authors need to do this for 3 samples. This is not a difficult task and requires 1-2 hours of time. Calculation of the average particle size should be carried out using the Sturges equation:

n = 1 + 3.22 × Log (N)

where N is the number of measured particles.

A1. We agree that more analysis methods and measurements is beneficial for every scientific paper. Considering the paper subject and objectives, we consider that the methods used are appropriate and the interpretation cover each chapter and subchapters. We understand that this is not enough for your criteria’s and we appreciate your help in improving the manuscript quality. Right now, our analysis methods are based mostly on our research infrastructure which don’t include the devices required in your evaluation.

Q2. High-impact research in the Q1 journal means using standard methods to analyse samples. The “Materials” just publishes articles where a qualitative analysis of the properties of materials is carried out. If the authors want to correspond a high level, it is necessary to do the most standard methods of analysis, such as XRF, ICP and LD.

A2. We acknowledge the advices regarding the criteria’s used in Q1 journal publications. Most of our papers are published in Q1 journals with IF varying from 3 to 24. However, the reviewer criteria may vary significantly and something it is not possible to provide a suitable answer to all demands. All your recommendations were very precious for us. Bellow we have included a short list of recent articles concerning zeolites which were published in “Materials” without XRD, ICP and LD.

Sun, H.; Lei, T.; Liu, J.; Guo, X.; Lv, J. Physicochemical Properties of Water-Based Copolymer and Zeolite Composite Sustained-Release Membrane Materials. Materials 2022, 15, 8553.

Dosa, M.; Grifasi, N.; Galletti, C.; Fino, D.; Piumetti, M. Natural Zeolite Clinoptilolite Application in Wastewater Treatment: Methylene Blue, Zinc and Cadmium Abatement Tests and Kinetic Studies. Materials 2022, 15, 8191.

Sheikh, A.; Akbari, M.; Shafabakhsh, G. Laboratory Study of the Effect of Zeolite and Cement Compound on the Unconfined Compressive Strength of a Stabilized Base Layer of Road Pavement. Materials 2022, 15, 7981.

Grabias-Blicharz, E.; Panek, R.; Franus, M.; Franus, W. Mechanochemically Assisted Coal Fly Ash Conversion into Zeolite. Materials 2022, 15, 7174.

Pabiś-Mazgaj, E.; Pichniarczyk, P.; Stempkowska, A.; Gawenda, T. Possibility of Using Natural Zeolite Waste Granules Obtained by Pressure Agglomeration as a Sorbent for Petroleum Substances from Paved Surfaces. Materials 2022, 15, 6871.

The authors
